# Approximated Backscattered Wave Models of a Lossy Concentric Dielectric Sphere for Fruit Characterization

Hoang Nam Dao 🔵, Chuwong Phongcharoenpanich 🔵 and Monai Krairiksh *

School of Engineering, King Mongkut's Institute of Technology Ladkrabang, Bangkok 10520, Thailand; namdh1986@gmail.com (H.N.D.); chuwong.ph@kmitl.ac.th (C.P.)
* Correspondence: monai.kr@kmitl.ac.th; Tel.: +66-818-281-831

**Abstract:** Approximated models of electromagnetic waves scattered from a sphere with two different dielectric layers were developed and reported in this paper. We proposed that the dielectric properties of a concentric dielectric sphere object, for example, some types of fruit, could be estimated by this model, from some wave components of the backscattered wave. The models were suitable for lossy objects because only a single bounce of the wave was assumed. In terms of first bounce as well as total backscattered wave results, the reported values agreed well with the values calculated by a commercial software. The measurement results verified the calculated wave components. The dielectric properties determination of real fruits was performed and exhibited the potential in fruit characterization. The main advantage of these models is that they can provide the magnitude and phase information of each backscattered wave component, which makes quality monitoring of fruits to be possible.

**Keywords:** scattered wave; concentric dielectric sphere; front axial return wave; rear axial return wave; glory wave; internal surface scattering wave; lossy dielectric

## 1. Introduction

In smart agricultural technology, non-destructive microwave sensors have been applied for classifying the quality of fruits such as tangerine (Citrus tangerine) [1], durian (*Durio* zibethinus) [2], and pomelo (Citrus maxima) [3]. These sensor techniques provided a good solution for post harvesting that can sense the quality of fruits off the trees. The key parameter of these works was the response of the wave from the fruit under test conditions, corresponding to different dielectric properties [4]. In general, the shape of the fruits such as tangerine, mangosteen (Garcinia mangos tana), melon (Cucumis melo), etc. can be approximated by a lossy two-layer concentric dielectric sphere. Mangosteen fruits, after harvesting, are classified into the export grade (large size–glossy peel, medium size–glossy peel, and large size–rough peel) and the domestic market grade (small size–glossy peel, medium size–rough peel, small size–rough peel, and undersize) based on the size and appearance. The major internal defect of mangosteen is translucent according to an excessive amount of water during mangosteen's development. The quality of mangosteen will decide its price. The appearance can be observed by human or machine, but the internal defect of mangosteen must be detected, non-destructively. One of the possible ways is to measure scattered wave from the fruit and determine its dielectric properties, which generally are different for normal and defected fruits. In this circumstance, it is necessary to develop a backscattered wave model for the lossy two-layer concentric dielectric sphere.

Several scattering wave models have been developed for the synthetic aperture radar imagery interpretation and the radar target recognition such as the Prony model [4–7], geometrical theory of diffraction model [8], attributed model [9,10], etc. The backscattered wave of a uniform plane wave by a dielectric sphere received the attention of many researchers according to the recognition of the dielectric objects, such as stealth-coated low-detectable

target and thermal-protective coated aircraft in radar application. The backscattered wave of the uniform plane wave by a homogeneous dielectric sphere was given in the form of an infinite series with a Mie solution, although it still has a limitation when the spherical radius exceeds a few wavelengths. This problem was overcome by utilizing the Watson transformation [11]. To investigate the wave components which contribute to the backscattered wave from the dielectric sphere, the modified geometrical optics method introduced by Thomas [12] was applied. It is apparent that the geometrical optic wave components, front axial return wave (FARW), rear axial return wave (RARW), and Glory wave (GW) contribute to the backscattered wave [13–15]. By comparing the total backscattered wave (summation of the FARW, RARW, and GW) with the backscattered wave given by the exact Mie solution, the final wave component contributed to the backscattered wave ISSW was pointed out in [16] and illustrated in [17]. A method involves the Watson transformation to split the exact Mie solution into the geometrical optics fields and the diffracted fields allows for the calculation of electric field intensity of each backscattered wave component [18,19]. Recently, the relative phases between these backscattered wave components were obtained by analyzing the ray path of each backscattered wave component for the low-loss dielectric spherical models as presented in [20].

The electric field intensity of the backscattered wave of the linear polarization plane wave by a multilayer lossy dielectric sphere has been widely investigated and presented in the form of a radar cross section of the lossy multi-layer dielectric sphere under the uniform plane wave [21–27]. However, the wave components of the backscattered wave, i.e., FARW, RARW, GW, and internal surface scattering wave (ISSW), which contribute to the backscattered wave, were not determined. The work on plane wave scattered by a core-shell sphere that wave components were presented and was firstly derived by Aden and Kerker [28]. However, calculation was rather time consuming according to multiple bounce of wave in the spherical object. This limits practical fruit classification, in which a large number of fruits and high speed is required. In the lossy media, the single bounce return wave is the main contribution of the backscattered wave since the multi-bounce rays are negligibly small according to high attenuation. It should be noted that the phase center of each backscattered wave component not only depends on their ray path but also on the loss factor of the spherical layer.

The objective of this work is to develop models for approximating backscattered waves from a lossy concentric dielectric sphere. While numerous research works on concentric dielectric spheres have been presented, the scattered wave was presented in terms of total scattered wave without showing the components of the wave. The wave components were presented only for the homogeneous dielectric sphere. The novelty of the research work in this paper is that the proposed models present wave components consist of FARW, RARW, GW, and ISSW for a lossy concentric dielectric sphere (LCS). These models provide information for investigating scattered wave from fruits which have spherical shape and particularly lossy dielectric. The knowledge from this investigation is useful for designing a sensor for classifying fruits with high speed. Since fruits consist of lossy dielectric properties for flesh and peel, in this work the backscattered wave models are considered to possess single bounce return wave from the lossy two-layer concentric dielectric sphere. The benefit of these models is that the wave components which directly relate to the dielectric properties of the inner layer can be determined and it is useful to characterize the quality of fruits non-destructively. Furthermore, the closed form expressions with single bounce provide a fast calculation.

This paper is organized as follows. Section 2 describes the background theory of the single bounce scattered wave models of the lossy two-layer concentric dielectric sphere. The calculation results that show the backscattered wave from the given two-layer concentric dielectric spherical models are described in Section 3. Section 4 shows an application of the proposed models for real fruit characterization. Finally, a conclusion is drawn in Section 5.

## 2. Materials and Methods

Since the problem of interest is to determine backscattered wave from a fruit with lossy dielectric properties, this section shows the approximated backscattered wave models in which only a single bounce was considered. The incident wave was a uniform plane wave polarized in the x direction that would hit a lossy concentric dielectric sphere (LCS) in normal, oblique, and grazing directions. To our best knowledge, the wave components of the backscattered wave from LCS have not been considered. In this section, the components of backscattered wave (FARW, RARW, GW, and ISSW) that explain the travelling phenomenon of these waves inside the LCS are presented. The major backscattered wave components from an incident wave hitting the LCS in the normal direction were front and rear axial return wave components, while those from an incident wave hitting the LCS in an oblique direction was a Glory backscattered wave component, and those from an incident wave hitting the LCS in a grazing direction was an internal surface scattering wave component. Through some calculations, the values of these properly measured wave components would yield the dielectric properties of the LCS.

### 2.1. Two-Layer Lossy Concentric Dielectric Sphere Structure

The structure of the LCS is a two-layer dielectric sphere structure characterized by the permittivity $(\varepsilon_1, \varepsilon_2)$ and permeability $(\mu_1, \mu_2)$ of each layer, see Figure 1. Since the scope of this study did not include magnetic material, the permeabilities of the LCS were considered as fixed constants, i.e., $\mu_0 = \mu_1 = \mu_2 = 4\pi \times 10^{-7}$H/m. In general, each layer of a lossy medium is specified by its relative complex permittivity $(\varepsilon_{ri}; i = 1, 2)$,

$$\varepsilon_{r_i} = \frac{\varepsilon_i}{\varepsilon_0} = \frac{\varepsilon'_i - j\varepsilon''_i}{\varepsilon_0} = \varepsilon_{r_i}' - j\varepsilon''_{r_i}, \qquad \text{for } i = 1, 2 \tag{1}$$

where $\varepsilon_0 = 8.85 \times 10^{-12}$ F/m is permittivity of vacuum. The conductivity $(\sigma_i; \ i = 1, 2)$ of each layer of LCS was

$$\sigma_i = \omega\varepsilon''_{r_i}\varepsilon_0, \text{ for } i = 1, 2 \tag{2}$$

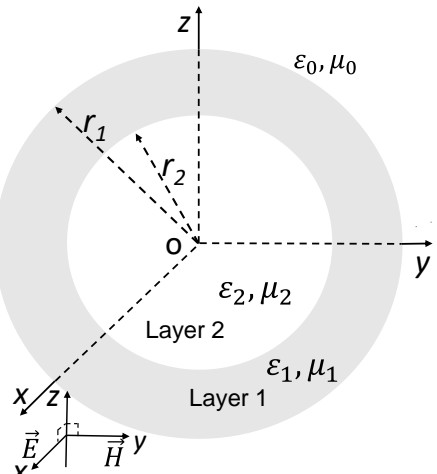

**Figure 1.** Geometry of the problem.

Established equations for determination of wave impedance, attenuation, and phase constant were taken from [29].

As the transmitted uniform plane wave propagated from the air to layer 1 and layer 2 of the LCS (with radii of $r_1$ and $r_2$), the transmitted and reflected wave must comply with Snell's law [29], where $\beta_0 = k = \omega\sqrt{\mu_0\varepsilon_0}$ is the phase constant of the uniform plane wave

propagating in the air. $n_0, n_1, n_2$ are the refractive indices of the air ($n_0 = 1$), layer 1, and layer 2, respectively. They are calculated by Equation (3) below,

$$n_i = \sqrt{\varepsilon'_{ri}} \left\{ \sqrt{1 + \left(\frac{\sigma_i}{\omega \varepsilon'_i}\right)^2} + 1 \right\}^{1/2}, \quad \text{for } i = 1, 2. \tag{3}$$

### 2.2. Front Axial Backscattered Wave Model

Front axial backscattered wave model is a model of front axial backscattered wave component from an LCS. Front axial backscattered wave will be present when the LCS has layers with different dielectric properties. The return wave consists of a single return wave from layer 1 ($F_1$) and a single return wave from layer 2 ($F_2$), as shown in Figure 2a. $F_1$ wave reflects back from the boundary of layer 1 in the reverse direction of normal incident wave. The equation for electric field intensity of $F_1$ wave is

$$E_{F_1} = \left( |\Gamma_{01}| e^{j\varphi_{\Gamma_{01}}} e^{-j2kr_1} \right) \left( \frac{r_1}{2} \frac{e^{-jkz}}{z} \right) \tag{4}$$

where $\Gamma_{01} = |\Gamma_{01}| e^{j\varphi_{\Gamma_{01}}}$ is the complex reflection coefficient (see Appendix A). $F_2$ wave travels through the boundary of layer 1 deep into the LCS and it is delayed compared to $F_1$. The delayed phase of the electric field of $F_2$ wave can be calculated from the different ray path length between $F_1$ and $F_2$. Therefore, the electric field intensity of the $F_2$ wave is given by

$$E_{F_2} = \left( T_{01} T_{10} \Gamma_{12} e^{-2\alpha_1(r_1 - r_2)} e^{-jk[2r_1 - 2n_1(r_1 - r_2)]} \right) \left( F_{F_2}(z) e^{-jkz} \right) \tag{5}$$

where $T_{01} = |T_{01}| e^{j\varphi_{T_{01}}}$ and $T_{10} = |T_{10}| e^{j\varphi_{T_{10}}}$ are the complex transmission coefficients at the boundary of layer 1; $\Gamma_{12} = |\Gamma_{12}| e^{j\varphi_{\Gamma_{12}}}$ is the complex reflection coefficient at the boundary of layer 2; and $F_{F_2}(z)$ is the spatial attenuation factor, calculated by the same procedure reported in [12]. The resulting equation for $F_{F_2}(z)$ is then $F_{F_2}(z) = \frac{n_1 r_2}{r_1 - r_2 + n_1 r_2} \frac{r_1}{2z}$.

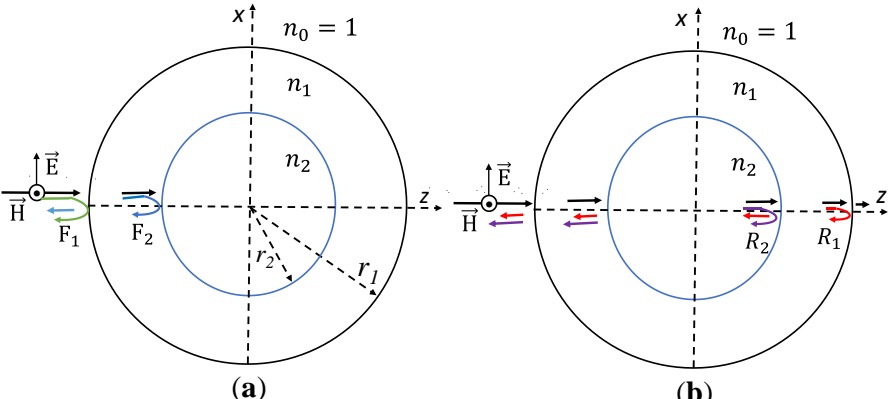

**Figure 2.** Backscattered waves from a normal incident wave on two-layer dielectric sphere: (**a**) Front axial return wave; (**b**) Rear axial single bounce return wave.

### 2.3. Rear Axial Backscattered Wave Model

Rear axial backscattered wave model is a model of the rear axial backscattered wave component. Figure 2b shows the propagation of the rear axial backscattered wave consisting of two single bounce return waves ($R_1$: outer sphere internal reflected wave and $R_2$: inner sphere internal reflected wave). $E_{R_1}$ travels into layer 1 and $4r_2$ deep into layer 2. It is

delayed compared to $E_{R_2}$. The delay can be calculated from the difference between the ray path lengths. The electric field intensity of $E_{R_1}$ is then expressed as follows,

$$E_{R_1} = \left( |\Gamma_{R_1}| E_{R_1}^{Att} e^{-jk[2r_1 - 4n_1(r_1 - r_2) - 4n_2 r_2] + j\varphi_{\Gamma_{R_1}}} \right) \left( F_{R_1}(z) e^{-jkz} \right) \qquad (6)$$

where $\Gamma_{R_1} = |T_{01}||T_{10}||T_{12}|^2|T_{21}|^2|\Gamma_{10}|, \varphi_{\Gamma_{R_1}} = \varphi_{T_{01}} + \varphi_{T_{10}} + 2\varphi_{T_{12}} + 2\varphi_{T_{21}} + \varphi_{\Gamma_{10}}$, and $T_{ml}|_{(m,l)=(0,1),(1,0)(1,2),(1,2)} = |T_{ml}| \exp(j\varphi_{T_{ml}}), (m,l) = (0,1),(1,0),(1,2),(2,1)$ are the refraction coefficients; $\Gamma_{10} = |\Gamma_{10}| \exp(\varphi_{\Gamma_{10}})$ is the reflection coefficient (see Appendix A); and $F_{R_1}(z)$ is the spatial attenuation factor, calculated by the same procedure reported in [12]. The resulting equation is:

$$F_{R_1}(z) = \frac{n_1 r_1 r_2 n_2^2}{(2n_1^2 - 3n_1 n_2 + n_2^2)r_1 + (4n_1^3 - 4n_1^2 + n_1 n_2 - n_2^2 + n_1 n_2^2 - 3n_1^2 n_2)r_2} \frac{1}{z}.$$

$E_{R_2}$ travels into layer 2 and it is delayed compared to $E_{F_1}$. The delay can be calculated from the difference between the ray path lengths. The electric field intensity of $R_2$ wave is obtained and expressed as follows

$$E_{R_2} = \left( \Gamma_{R_2} e^{j\varphi_{\Gamma_{R_2}}} e^{-2\alpha_1(r_1 - r_2) - 4\alpha_2 r_2} e^{-jk[2r_1 - 2n_1(r_1 - r_2) + 4n_2 r_2]} \right) \left( F_{R_2}(z) e^{-jkz} \right) \qquad (7)$$

where $\Gamma_{R_2} = |T_{01}||T_{10}||T_{12}||T_{21}||\Gamma_{21}|$, $\varphi_{\Gamma_{R_2}} = \varphi_{T_{01}} + \varphi_{T_{10}} + \varphi_{T_{12}} + \varphi_{T_{21}} + \varphi_{\Gamma_{21}}$;

$T_{ml}|_{(m,l)=(0,1),(1,0)(1,2),(1,2)} = |T_{ml}| \exp(j\varphi_{T_{ml}}), (m,l) = (0,1),(1,0),(1,2),(2,1)$ are the refraction coefficients; $\Gamma_{21} = |\Gamma_{21}| \exp(\varphi_{\Gamma_{21}})$ is the reflection coefficient (see Appendix A); and $F_{R_2}(z)$ is the spatial attenuation factor, calculated by the same procedure reported in [12]. The resulting equation for $F_{R_2}(z)$ is then $F_{R_2}(z) = \frac{n_1 n_2 r_2}{(-2n_1 + n_2)r_1 + (n_1 - 1)n_2 r_2} \frac{r_1}{2z}$.

### 2.4. Glory Backscattered Wave Model

The Glory backscattered wave model (GW) is a model of backscattered wave component from an incident wave hitting the LCS in an oblique direction. For the LCS defined in this study, the presence of the Glory wave component not only depended upon the dielectric constants $(\varepsilon_1, \varepsilon_2)$, and the angle of oblique incident wave of the dielectric sphere as reported in [12–15] but it also depended upon the dimension $(r_1, r_2)$. The magnitude of the Glory wave can be calculated from the spherical dimension $(r_1, r_2)$, the attenuation constant of each spherical layer $(\alpha_1, \alpha_2)$, and the transmission and reflection coefficients of the Glory wave at the boundary of LCS. The phase center of the Glory wave is a function of the ray path of the wave travelling inside the sphere, which in turn, depended upon the refractive index $(n_1, n_2)$ and the dimension $(r_1, r_2)$ of the LCS.

Figure 3 shows the travelling paths of wave in the GW model with $r_1 = \sin\theta_1^i/h$, where $h$ is the distance from the origin of the LCS to the ray path of the refracted wave into the LCS for four conditions and a fixed $r_2$ as shown in Figure 3a–d. The Glory wave can be presented with four cases according to incident angle, dimension, and dielectric properties of the LCS.

Case 1: $h = \frac{r_1 \sin\theta_1^i}{n_1} > r_2$.

The oblique incident wave enters layer 1 with refraction coefficients $T_{\|01}$ and $T_{\perp 01}$ at a refraction angle $\theta_1^t = \sin^{-1}(\sin\theta_1^i/n_1)$ (Snell's law of refraction) and travels into layer 1. Before the wave emerges from the LCS with transmission coefficients $T_{\|10}$ and $T_{\perp 10}$ in Figure 3, it reflected at the boundary of layer 1 $\left( \Gamma_{\|10}, \Gamma_{\perp 10} \right)$ as shown in Figure 3a. The Glory wave emerged from the LCS is the backscattered wave, as shown in Figure 3a.

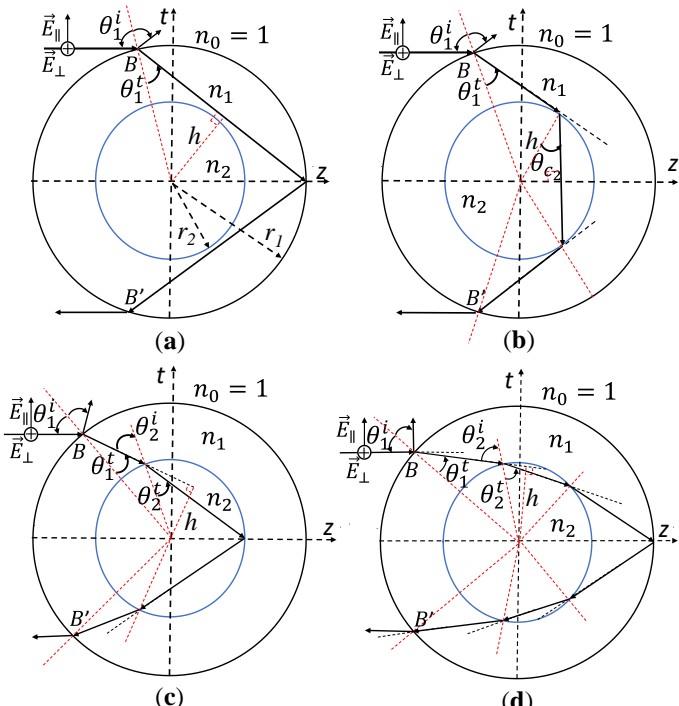

**Figure 3.** Backscattered waves of the oblique incident wave from LCS for: (a) $\begin{cases} h = \frac{r_1 \sin \theta_1^i}{n_1} > r_2 \\ \theta_1^i = 2\theta_1^t \end{cases}$ ;

(b) $\begin{cases} h = \frac{r_1 \sin \theta_1^i}{n_1} = r_2 \\ \theta_1^i = \theta_1^t + \theta_{c_2} \end{cases}$ ; (c) $\begin{cases} h = \frac{r_1 \sin \theta_1^i}{n_1} < r_2 \\ \theta_1^i + \theta_2^i = \theta_1^t + 2\theta_2^t \end{cases}$ ; (d) $\begin{cases} h = \frac{r_1 \sin \theta_1^i}{n_1} < r_2 \\ \theta_1^i + 2\theta_2^i = 2\theta_1^t + 2\theta_2^t \end{cases}$ .

The Glory wave enters and emerges from the LCS at the points B and B' on the circumference, respectively. The electric field intensity of the Glory wave polarized in x direction in lossy media can be calculated as follows

$$E_{G_1}^s = \left( E_{0G_1} e^{-\alpha_1 2 n_1 r_1} e^{j\varphi_{G_{1x}}} \right) \left( \frac{\pi \sqrt{r_1^3} n_1 \sqrt{(8 - 3n_1^2)(n_1^2 - 2)}}{2\sqrt{\lambda}} \frac{e^{-jkz}}{z} \right) \tag{8}$$

$$\begin{cases} E_{0G_1} = \frac{1}{2} \sqrt{\left( \left| \Gamma_{\parallel G_1} \right| \cos \varphi_{\parallel G_1} - \left| \Gamma_{\perp G_1} \right| \cos \varphi_{\perp G_1} \right)^2 + \left( \left| \Gamma_{\parallel G_1} \right| \sin \varphi_{\parallel G_1} - \left| \Gamma_{\perp G_1} \right| \sin \varphi_{\perp G_1} \right)^2} \\ \varphi_{G_{1x}} = -k(2r_1 \cos \theta_1^i - 2n_1^2 r_1) + \tan^{-1} \left( \frac{\left| \Gamma_{\parallel G_1} \right| \sin \varphi_{\parallel G_1} - \left| \Gamma_{\perp G_1} \right| \sin \varphi_{\perp G_1}}{\left| \Gamma_{\parallel G_1} \right| \cos \varphi_{\parallel G_1} - \left| \Gamma_{\perp G_1} \right| \cos \varphi_{\perp G_1}} \right) \end{cases} \tag{9}$$

where $\left| \Gamma_{\parallel G_1} \right| = \left| T_{\parallel 01} \right| \left| T_{\parallel 10} \right| \left| \Gamma_{\parallel 10} \right|$; $\left| \Gamma_{\perp G_1} \right| = \left| T_{\perp 01} \right| \left| T_{\perp 10} \right| \left| \Gamma_{\perp 10} \right|$; $\varphi_{\parallel G_1} = \varphi_{T_{\parallel 10}} + \varphi_{T_{\parallel 01}} + \varphi_{\Gamma_{\parallel 10}}$; $\varphi_{\perp G_1} = \varphi_{T_{\perp 01}} + \varphi_{T_{\perp 10}} + \varphi_{\Gamma_{\perp 10}}$; and $\left. T_{\parallel ml} \right|_{(m,l)=(0,1),(1,0)} = \left| T_{\parallel ml} \right| \exp\left( j\varphi_{\parallel ml} \right)$; $\Gamma_{\parallel 10} = \left| \Gamma_{\parallel 10} \right| \exp\left( j\varphi_{\parallel 10} \right)$; $\left. T_{\perp ml} \right|_{(m,l)=(0,1),(1,0)} = \left| T_{\perp ml} \right| \exp\left( j\varphi_{\perp ml} \right)$; $\Gamma_{\perp 10} = \left| \Gamma_{\parallel 10} \right| \exp\left( j\varphi_{\perp 10} \right)$; see the derivations in the Appendix A.

Case 2: $h = \frac{r_1 \sin \theta_1^i}{n_1} = r_2$.

The oblique incident wave enters layer 1 of the LCS as the above description in case 1. The refraction coefficients at the boundary of layer 1 are $T_{\parallel 01}$, $T_{\perp 01}$ (for entering) and $T_{\parallel 10}$, $T_{\perp 10}$ (for emerging). Inside layer 1, Glory wave enters and exits the inner sphere at the grazing direction as depicted in Figure 3b. The electric field of Glory wave polarized in lossy media in the x-direction is obtained as follows

$$E_{G_2}^s = \left( E_{0G_2} e^{-2\alpha_1 r_1 \sqrt{n_1^2 - \sin^2 \theta_1^i} - 2\alpha_2 r_2 \sqrt{n_2^2 - n_1^2}} e^{j\varphi_{G_2}} \right) \left( \frac{\sqrt{2}\pi r_1^{3/2} \sin \theta_1^i}{\sqrt{\lambda}} \sqrt{\cos \theta_1^i \frac{n_1 \cos \theta_1^t - 2 \cos \theta_1^i}{n_1 \cos \theta_1^t - \cos \theta_1^i}} \frac{e^{-jkz}}{z} \right) \tag{10}$$

$$\begin{cases} E_{0G_2} = \frac{1}{2} \sqrt{\left( \left| T_{\parallel_{G_2}} \right| \cos \varphi_{\parallel_{G_2}} - \left| T_{\perp_{G_2}} \right| \cos \varphi_{\perp_{G_2}} \right)^2 + \left( \left| T_{\parallel_{G_2}} \right| \sin \varphi_{\parallel_{G_2}} - \left| T_{\perp_{G_2}} \right| \sin \varphi_{\perp_{G_2}} \right)^2} \\ \varphi_{G_2} = -k \left( 2r_1 \cos \theta_1^i - L_{G_2\_delay} \right) + \tan^{-1} \left( \frac{\left| T_{\parallel_{G_2}} \right| \sin \varphi_{\parallel_{G_2}} - \left| T_{\perp_{G_2}} \right| \sin \varphi_{\perp_{G_2}}}{\left| T_{\parallel_{G_2}} \right| \cos \varphi_{\parallel_{G_2}} - \left| T_{\perp_{G_2}} \right| \cos \varphi_{\perp_{G_2}}} \right) \end{cases} \tag{11}$$

Case 3: $h = \frac{r_1 \sin \theta_1^i}{n_1} < r_2$ and $\theta_1^i + \theta_2^i = \theta_1^t + 2\theta_2^t$.

In Figure 3c, the wave reflects at the boundary of layer 2 with reflection coefficients $\left( \Gamma_{\parallel 21}, \Gamma_{\perp 21} \right)$ at an angle, $\theta_2^t$, and emerges from the inner sphere at a refraction angle, $\theta_2^i$, with refraction coefficients $\left( T_{\parallel 21}, T_{\perp 21} \right)$. The Glory wave in this case is shown below.

$$E_{G_3}^s = \left( E_{0G_3} E_{G_3\_att} e^{j\varphi_{G3}} \right) \left( \frac{\sqrt{2}\pi r_1^{3/2} \sin \theta_1^i}{\sqrt{\lambda}} \sqrt{\frac{1 + \frac{2r_1 \cos \theta_1^i}{n_1 r_2 \cos \theta_2^i} - \frac{2 \cos \theta_1^i}{n_1 \cos \theta_1^t} - \frac{4r_1 \cos \theta_1^i}{n_2 r_2 \cos \theta_2^i}}{1 + \frac{r_1 \cos \theta_1^i}{n_1 r_2 \cos \theta_2^i} - \frac{\cos \theta_1^i}{n_1 \cos \theta_1^t} - \frac{2r_1 \cos \theta_1^i}{n_2 r_2 \cos \theta_2^i}}} \cos \theta_1^i \frac{e^{-jkz}}{z} \right) \tag{12}$$

$$\begin{cases} E_{0G_3}^s = \frac{1}{2} \sqrt{\left( \left| \Gamma_{\parallel_{G_3}} \right| \cos \varphi_{\parallel_{G_3}} - \left| \Gamma_{\perp_{G_3}} \right| \cos \varphi_{\perp_{G_3}} \right)^2 + \left( \left| \Gamma_{\parallel_{G_3}} \right| \sin \varphi_{\parallel_{G_3}} - \left| \Gamma_{\perp_{G_3}} \right| \sin \varphi_{\perp_{G_3}} \right)^2} \\ \varphi_{G_3} = -kL_{3\_delay} + \tan^{-1} \left( \frac{\left| \Gamma_{\parallel_{G_3}} \right| \sin \varphi_{\parallel_{G_3}} - \left| \Gamma_{\perp_{G_3}} \right| \sin \varphi_{\perp_{G_3}}}{\left| \Gamma_{\parallel_{G_3}} \right| \cos \varphi_{\parallel_{G_3}} - \left| \Gamma_{\perp_{G_3}} \right| \cos \varphi_{\perp_{G_3}}} \right) \end{cases} \tag{13}$$

where $\varphi_{\parallel_{G_3}} = \varphi_{T_{\parallel 01}} + \varphi_{T_{\parallel 10}} + \varphi_{T_{\parallel 12}} + \varphi_{T_{\parallel 21}} + \varphi_{\Gamma_{\parallel 21}}$, $\varphi_{\perp_{G_3}} = \varphi_{T_{\perp 01}} + \varphi_{T_{\perp 10}} + \varphi_{T_{\perp 12}} + \varphi_{T_{\perp 21}} + \varphi_{\Gamma_{\perp 21}}$, $\left| \Gamma_{\parallel_{G_3}} \right| = \left| T_{\parallel 01} \right| \left| T_{\parallel 10} \right| \left| T_{\parallel 12} \right| \left| T_{\parallel 21} \right| \left| \Gamma_{\parallel 21} \right|$, $\left| \Gamma_{\perp_{G_3}} \right| = |T_{\perp 01}||T_{\perp 10}||T_{\perp 12}||T_{\perp 21}||\Gamma_{\perp 21}|$.

Case 4: $h = \frac{r_1 \sin \theta_1^i}{n_1} = r_2$, and $\theta_1^i + 2\theta_2^i = 2\theta_1^t + 2\theta_2^t$.

In Figure 3d, after hitting layer 2 boundary, the wave transmits into layer 1 with the refraction coefficient $\left( T_{\parallel 21}, T_{\perp 21} \right)$ at the refraction angle, $\theta_2^i$. This wave reflects at layer 1 boundary with the reflection coefficient $\left( \Gamma_{\parallel 10}, \Gamma_{\perp 10} \right)$ at the reflection angle, $\theta_1^t$. The wave travels into layer 1 and enters layer 2 again with the refraction coefficient $\left( T_{\parallel 12}, T_{\perp 12} \right)$ at the refraction angle, $\theta_2^t$. The wave travels into layer 2, hits the layer 2 boundary, and returns to layer 1 with the refraction coefficients $\left( T_{\parallel 21}, T_{\perp 21} \right)$ at the refraction angle $\theta_2^i$. This wave emerges the two-layer dielectric sphere as the component of the backscattered wave with the refraction coefficient $\left( T_{\parallel 10}, T_{\perp 10} \right)$ at the refraction angle, $\theta_1^i$.

The electric field intensity of the Glory wave polarized in this case is obtained as follows

$$E_{G_4}^s = \left( E_{0G_4} E_{G_4\_att} e^{j\varphi_4} \right) \left( \frac{\sqrt{2}\pi r_1^{3/2} \sin \theta_1^i}{\sqrt{\lambda}} \sqrt{\frac{1 + \frac{2r_1 \cos \theta_1^i}{n_1 r_2 \cos \theta_2^i} - \frac{2 \cos \theta_1^i}{n_1 \cos \theta_1^t} - \frac{4r_1 \cos \theta_1^i}{n_2 r_2 \cos \theta_2^i}}{1 + \frac{r_1 \cos \theta_1^i}{n_1 r_2 \cos \theta_2^i} - \frac{\cos \theta_1^i}{n_1 \cos \theta_1^t} - \frac{2r_1 \cos \theta_1^i}{n_2 r_2 \cos \theta_2^i}}} \cos \theta_1^i \frac{e^{-jkz}}{z} \right) \tag{14}$$

$$
\left\{
\begin{aligned}
E_{0G_4}^s &= \tfrac{1}{2} \sqrt{ \left( \left| \Gamma_{\|G_4} \right| \cos \varphi_{\|G_4} - \left| \Gamma_{\perp G_4} \right| \cos \varphi_{\perp G_4} \right)^2 + \left( \left| \Gamma_{\|G_4} \right| \sin \varphi_{\|G_4} - \left| \Gamma_{\perp G_4} \right| \sin \varphi_{\perp G_4} \right)^2 } \\
\varphi_{G_4} &= -kL_{4\_delay} + \tan^{-1}\left( \frac{\left| \Gamma_{\|G_3} \right| \sin \varphi_{\|G_4} - \left| \Gamma_{\perp G_4} \right| \sin \varphi_{\perp G_4}}{\left| \Gamma_{\|G_4} \right| \cos \varphi_{\|G_4} - \left| \Gamma_{\perp G_4} \right| \cos \varphi_{\perp G_4}} \right)
\end{aligned}
\right.
\tag{15}
$$

where $\varphi_{\|G_4} = \varphi_{T_{\|01}} + \varphi_{T_{\|10}} + 2\varphi_{T_{\|12}} + 2\varphi_{T_{\|21}} + \varphi_{\Gamma_{\|10}}$, $\varphi_{\perp G_4} = \varphi_{T_{\perp 01}} + \varphi_{T_{\perp 10}} + \varphi_{T_{\perp 12}} + 2\varphi_{T_{\perp 21}} + 2\varphi_{\Gamma_{\perp 21}} + \varphi_{\Gamma_{\perp 10}}$ $\left| \Gamma_{\|G_4} \right| = \left| T_{\|01} \right| \left| T_{\|10} \right| \left( \left| T_{\|12} \right| \left| T_{\|21} \right| \right)^2 \left| \Gamma_{\|10} \right|$, $\left| \Gamma_{\perp G_4} \right| = \left| T_{\perp 01} \right| \left| T_{\perp 10} \right|$ $\left( \left| T_{\perp 12} \right| \left| T_{\perp 21} \right| \right)^2 \left| \Gamma_{\perp 10} \right|$.

It should be noted that for $n_1 > n_2$, the Glory wave has the same physical phenomenon of the wave travelling inside the two-layer dielectric sphere as the above description in Figure 3c,d with $\theta_2^i < \theta_2^t$.

### 2.5. Internal Surface Scattering Wave Model

The internal surface scattering wave (ISSW) exists with a unique characteristic of the dielectric sphere as considered in the homogeneous dielectric sphere [12–15]. The grazing incident wave takes the shortcut inside the dielectric sphere before becoming the surface wave at the sphere shadow. It travels on the spherical surface and emerges in the backscattered wave at the conjugate points (as C, C′). The attenuation factors of electric field inside layer 1, layer 2, and on the surface of the dielectric sphere depending on the attenuation constants, $\alpha_1, \alpha_2$, and $\alpha_s$, respectively.

The ISSW enters layer 1 at the points with the critical angle $\theta_{c_1} = \sin^{-1}(1/n_1)$. The travelling of the ISSW in layer 1 has the same procedure as the above description and the three ISSW models are obtained as depicted in Figure 4.

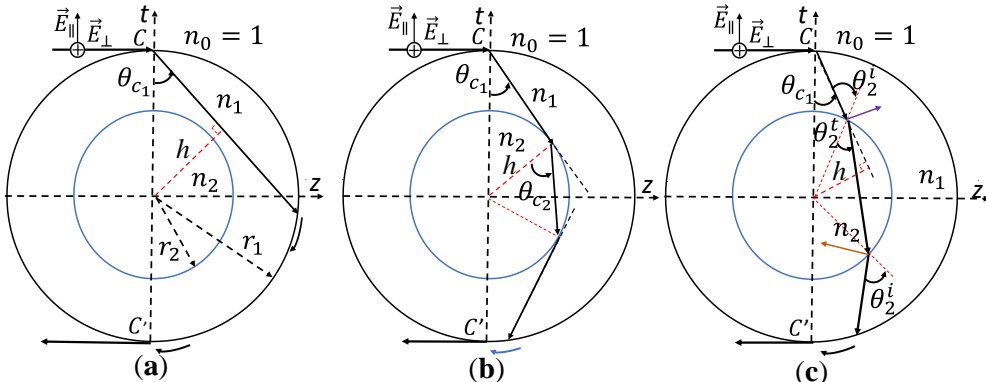

**Figure 4.** Backscattered waves of two-layer dielectric sphere illuminated by the grazing incident wave: (a) $h = \frac{r_1}{n_1} > r_2$; (b) $\left\{ \begin{array}{l} h = \frac{r_1}{n_1} = r_2 \\ \theta_{c_1} + \theta_{c_2} \geq \pi/2 \end{array} \right.$; (c) $\left\{ \begin{array}{l} h = \frac{r_1}{n_1} < r_2 \\ \theta_2^i \leq \theta_{c_1} + \theta_2^t \end{array} \right.$.

The ISSW can be presented in three cases as follows

Case 1: $h = \frac{r_1}{n_1} > r_2$.

The ISSW takes a shortcut in layer 1 and appears as the surface wave. This surface wave travels on the spherical surface before emerging in the backscattered wave as shown in Figure 4a. The electric field intensity of this ISSW is obtained as follows

$$
E_{I_1}^s = D_1 e^{-\frac{2\alpha_1 r_1}{n_1} \sqrt{n_1^2 - 1}} e^{-\alpha_s (\pi - 2\theta_{c1}) r_1} e^{j2k(r_1 \sqrt{n_1^2 - 1} + \theta_{c1} r_1)} \frac{e^{-jkz}}{z}
\tag{16}
$$

where $D_1$ is spatial attenuation factor that depends upon the diameter of the sphere and the refractive index of the medium.

Case 2: $h = \frac{r_1}{n_1} = r_2$.

The ISSW enters and exits layer 1 before emerging in the backscattered wave with the same procedure as case 1. The ISSW travels in layer 1 and enters layer 2 at the grazing direction of the inner sphere. It travels in layer 2 before emerging in the grazing direction of the inner sphere. The ISSW electric field intensity is given by

$$E_{I_2}^s = D_2 e^{-\frac{2\alpha_1 r_1}{n_1}\sqrt{n_1^2-1}-\frac{2\alpha_2 r_2}{n_2}\sqrt{n_2^2-n_1^2}} e^{-\alpha_s 2r_1[\sin^{-1}(\frac{1}{n_1})+\sin^{-1}(\frac{n_1}{n_2})-\frac{\pi}{2}]} e^{-jkL_{I_2\_delay}} \frac{e^{-jkz}}{z} \quad (17)$$

where $D_2$ is spatial attenuation factor that depends upon the diameter of the sphere and the refractive index of the medium.

Case 3: $h = \frac{r_1}{n_1} < r_2$.

In this case, after entering layer 1, the ISSW hits layer 2 boundary and transmits into layer 2. It travels inside layer 2 before hitting layer 2 boundary. The ISSW transmits into layer 1 at the refraction angle $\theta_2^i$ and travels in layer 1 before appears as the surface wave that travels on the outer spherical surface as depicted in Figure 4c. To obtain the ISSW emerging from the two-layer dielectric sphere in the backscattered wave, the emerging point of the surface wave must be in the shadow area of the dielectric. The electric field intensity of ISSW in the lossy media is

$$E_{I_3}^s = D_3 E_{0I_3}^s \exp\left[-2\alpha_1\left(\frac{r_1}{n_1}\sqrt{n_1^2-1} - \frac{1}{n_1}\sqrt{n_1^2 r_2^2 - r_1^2}\right) - \frac{2\alpha_2}{n_2}\sqrt{n_2^2 r_2^2 - r_1^2}\right]$$
$$\times \exp\left[-\alpha_s 2r_1\left[\sin^{-1}(1/n_1) + \sin^{-1}(r_1/n_2 r_2) - \sin^{-1}(r_1/n_1 r_2)\right]\right] e^{j\varphi_{G_3}} \frac{e^{-jkz}}{z} \quad (18)$$

where $D_3$ is spatial attenuation factor that depends upon the diameter of the sphere and the refractive index of the medium.

$$\begin{cases} E_{0I_3}^s = \frac{1}{2}\sqrt{\left(\left|T_{\|I_3}\right|\cos\varphi_{\|I3} - \left|T_{\perp I_3}\right|\cos\varphi_{\perp I3}\right)^2 + \left(\left|T_{\|I_3}\right|\sin\varphi_{\|I3} - \left|T_{\perp I_3}\right|\sin\varphi_{\perp I3}\right)^2} \\ \\ \varphi_{G_3} = -kL_{I_{3delay}} + \tan^{-1}\left(\dfrac{\left|T_{\|I_3}\right|\sin\varphi_{\|I_3} - \left|T_{\perp I_3}\right|\sin\varphi_{\perp I_3}}{\left|T_{\|I_3}\right|\cos\varphi_{\|I_3} - \left|T_{\perp I_3}\right|\cos\varphi_{\perp I_3}}\right) \end{cases} \quad (19)$$

where $\left|T_{\|I_3}\right| = \left|T_{\|12}\right|\left|T_{\|21}\right|$, $\left|T_{\perp I_3}\right| = |T_{\perp 12}||T_{\perp 21}|$, $\varphi_{\|I_3} = \varphi_{T_{\|12}} + \varphi_{T_{\|21}}$, $\varphi_{\perp I_3} = \varphi_{T_{\perp 12}} + \varphi_{T_{\perp 21}}$, see Appendix A.

$$L_{I_{3delay}} = 2\left(r_1\sqrt{n_1^2-1} - \sqrt{n_1^2 r_2^2 - r_1^2}\right) + 2\sqrt{n_2^2 r_2^2 - r_1^2} + 2r_1\left[\sin^{-1}\left(\frac{1}{n_1}\right) + \sin^{-1}\left(\frac{r_1}{n_2 r_2}\right) - \sin^{-1}\left(\frac{r_1}{n_1 r_2}\right)\right].$$

## 3. Results

In this section, four LCS structures (LCS 1, 2, 3, and 4) were used to calculate the above backscattered wave models. The electric field intensities and the phase centers of each backscattered wave component were calculated. The total backscattered wave for LCS 1 and 2 were compared with those calculated with CST Studio Suite (CST) [30]. Since the comparison could be performed for only the total electric field intensity, to verify the components of the wave, two experiments were setup for the two LCSs to determine the magnitude and phase center of wave components for LCS 3 and 4.

### 3.1. LCS 1 ($r_1 = 3\,cm$, $r_2 = 1.7\,cm$, $\varepsilon_{r_0} = 1$, $\varepsilon_{r_1} = 2.59 + 0.05j$, $\varepsilon_{r_2} = 4.84 + 0.1j$, $f = 10\,GHz$)

This model had quite low loss and the backscattered wave consisted of seven wave components, i.e., $F_1$, $F_2$, $R_1$, $R_2$, $I_1$, $G_1$, and $G_3$ as shown in Figure 5a. The internal surface scattering wave did not cross the inner dielectric sphere. Therefore, $I_1$ wave was an agent of the internal surface scattering wave. The Glory wave had two components ($G_1$ and $G_3$ waves) for the oblique incident wave angles (72.8° and 43°). For the normal incident wave, the front axial return wave ($F_1$ and $F_2$ waves) and the rear axial return

wave ($R_1$ and $R_2$ waves) exist and contribute to the backscattered wave. Figure 5b shows the magnitude of scattered wave components versus phase center displayed in terms of equivalent diameter ($2n_1r_1$). The magnitude of the $I_1$ wave was the largest component while the $G_3$ wave was the smallest component. The large magnitude difference between $I_1$ wave and $G_3$ wave was due to their ray paths travelling inside the LCS (17.6 cm of $G_3$ wave and 7.6 cm of $I_1$ wave) and their incident waves (grazing incident wave of $I_1$ wave and oblique incident wave of $G_3$ wave). The magnitude of $G_1$ wave was greater than the one of $G_3$ wave although the ray path of $G_1$ wave was slightly longer than the one of $G_3$ wave since the $G_1$ wave travelled in layer 1 while the $G_3$ wave travels in both layer 1 and layer 2. For the rear axial return wave ($R_1$ and $R_2$ waves), the magnitude of $R_1$ wave was less than the one of $R_2$ wave although the higher reflection coefficient was at the layer 1 boundary compared to the one at layer 2. This came from the fact that the ray path of the $R_1$ wave was 2.6 cm longer than the one of $R_2$ wave, and the spatial attenuation factor of $R_1$ wave was 0.38 times the one of $R_2$ wave. For the front axial return wave, the $F_1$ wave had the earliest phase compared to other components and its magnitude depends upon the refractive index $n_1$ and the radius of the outer sphere. The field of $F_2$ wave was smaller than the one of the $F_1$ wave as shown in Figure 5b. It can be explained by the travelling of the $F_2$ wave in the low loss layer 1 and the higher reflection coefficient at layer 1 boundary than at layer 2 boundary. Compared with the total field of the backscattered wave simulated with CST [30], it was found that the error of the total field calculation was 18% for magnitude and 5.5% for phase center, respectively. The calculation error could be attributed from the significant effect from multi-bounce components ($F_2$, $G_1$ and $I_1$ waves) that exist in the low-loss dielectric sphere.

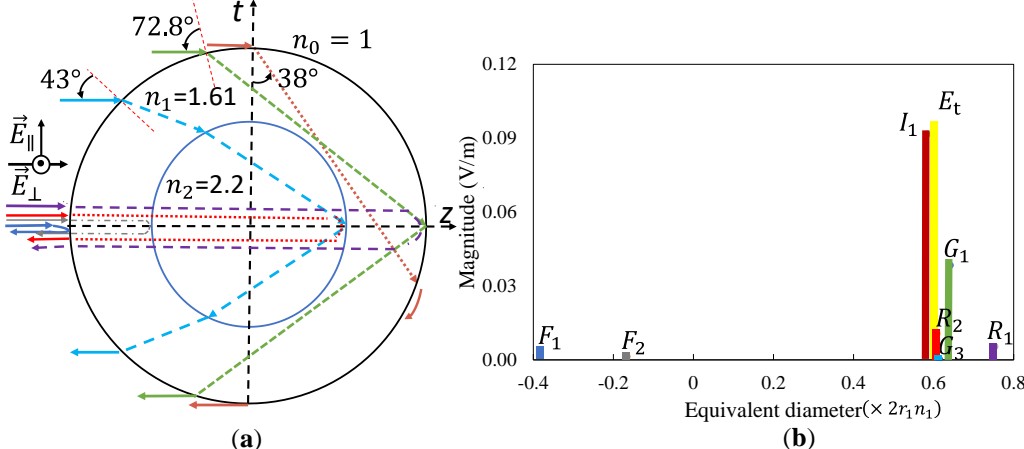

**(a)**          **(b)**

**Figure 5.** Backscattered wave for LCS 1: (**a**) Propagation of the backscattered wave components; (**b**) Magnitude and phase center of the backscattered wave components.

*3.2. LCS 2 ($r_1 = 3\ cm, r_2 = 1.7\ cm, \varepsilon_{r_0} = 1, \varepsilon_{r_1} = 2.59 + 0.5j, \varepsilon_{r_2} = 9.61 + 1.6j, f = 10\ GHz$)*

In this case the relative dielectric constants of the outer and inner sphere were changed to $\varepsilon_{r_1} = 2.56 + 0.5j$, $\varepsilon_{r_2} = 9.61 + 1.6j$ while its geometry was same as the previous model. The backscattered wave components were shown in Figure 6a. The $G_2$ wave appeared when the oblique incident angle was 66.4°. Its ray path travelled a distance of 4.9 cm in layer 1 and took a shortcut of 2.9 cm in layer 2, and its ray path was 1.9 cm shorter than the one of $G_1$ wave (travelling in layer 1). The electric field intensity of $G_2$ wave was attenuated by 28.5 dB that was almost same as the one of $G_1$ wave (28.3 dB). The greater field of $G_1$ wave compared to the one of $G_2$ wave due to both the spatial attenuation factor and the transmitted and reflected coefficients. In Figure 6b, it was clear that $I_1$, $G_1$, $F_1$, and $F_1$ waves were the main contributors to the total field of the backscattered wave while $G_2$, $R_1$, $R_2$ waves were small and could be negligible. The total electric field intensity of the backscattered wave was very well matched with the one calculated

by CST [30]. The error for magnitude and phase center were 3.5% and 2%, respectively. The accuracy of this model was improved because the effect of the multi-bounce wave components was removed in the lossy dielectric sphere.

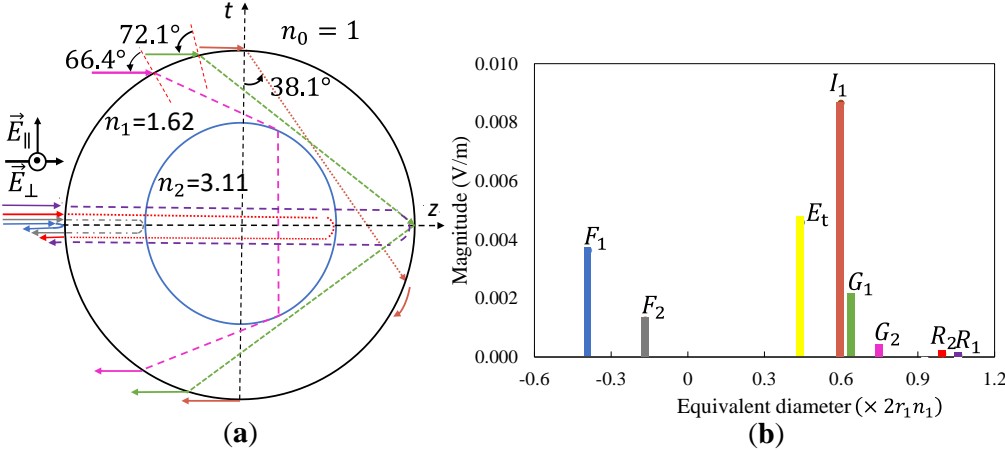

**Figure 6.** Backscattered wave for LCS 2: (**a**) Propagation of the backscattered wave components; (**b**) Magnitude and phase center of the backscattered wave components.

To verify the calculation results of wave components, experiments were setup to measure wave components. Without a time-domain measurement option of the available equipment, the frequency domain response of the backscattered wave from a lossy concentric dielectric sphere (LCS) consisted of two different dielectric layers. The backscattered wave from the LCS was measured by using an N9916A FieldFox network analyzer in the frequency range of 2.5–10.5 GHz with 1001 sample points. Note that the system was calibrated with an open, short, load at the two ports of the network analyzer. Then, the thru calibration was performed with the broadband antennas connected at the two ports of the network analyzer when they were pointed at a conducting plane. Substituting the conducting plane by a spherical model, the frequency domain response of backscattered wave was measured. Then, Fourier transform was taken to obtain time-domain response. The resolution of time response of system was 0.125 ns (corresponding to 3.75 cm distance of wave in air).

The backscattered wave components were collected from the LCS by a system consisted of an N9916A FieldFox network analyzer and an 83059A Agilent microwave amplifier as depicted in Figure 7a. The two antennas were placed closed to each other and separated by an RF absorber. The separation was small, and the monostatic radar could be demonstrated. The $S_{21}$ parameter of the air (without a conducting plane), conducting plane (Figure 7b and two LCS structures Figure 7c,d) were recorded. The backscattered wave was calculated as below

$$\left| S_{21}^s - S_{21}^A \right| \frac{4\pi d}{\lambda G} = \frac{\left| E^s \right|}{\left| E^i \right|}, \tag{20}$$

where $S_{21}^s, S_{21}^A$ were obtained from the measurement results of the sample (conducting plane, two LCSs) and the air measurement, respectively; $d$ is the distance between the phase center of the antenna to the surface of the object under test (conducting plane, two LCS) $d = R - 2.5 = 27.5$ cm; $G$ is the gain of transmitting and receiving antennas which have the same value. Note that the system was calibrated with an open, short, load at the two ports of the network analyzer without amplifier. The isolation between two ports was less than $-18$ dB for the whole frequency band.

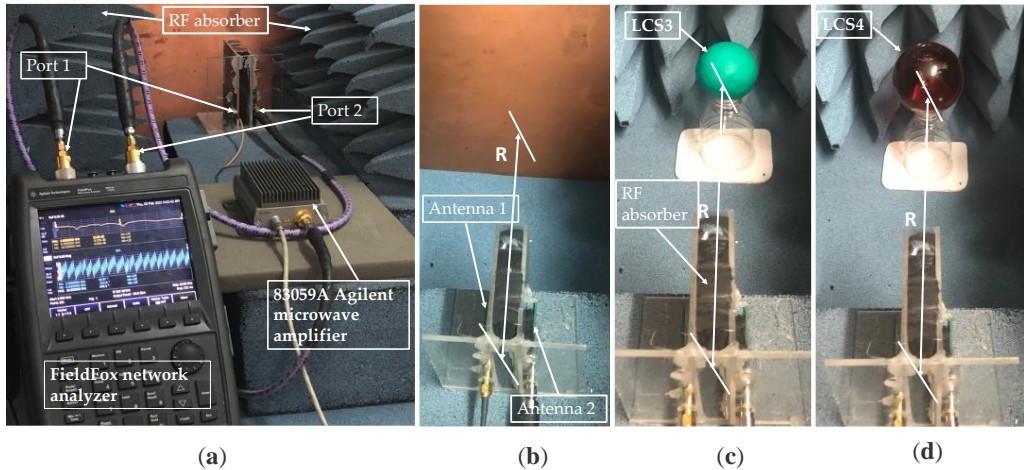

(a)    (b)    (c)    (d)

**Figure 7.** Measurement setup: (**a**) Whole measurement system; (**b**) Conducting plane measurement; (**c**) LCS 3 measurement; (**d**) LCS 4 measurement.

### 3.3. Conducting Plane Measurement Results

The time-domain response of the scattered wave from the conducting plane is displayed in Figure 8. In this presentation, we used the distance $(s = v \times t)$ on the horizontal axis instead of time to compare with the physical distance, where $v$ is the velocity of the wave in air, $v = 3 \times 10^8$ m/s. The vertical axis shows the ratio of the scattered electric field intensity $(E_s)$ to incident electric field intensity $(E_i)$.

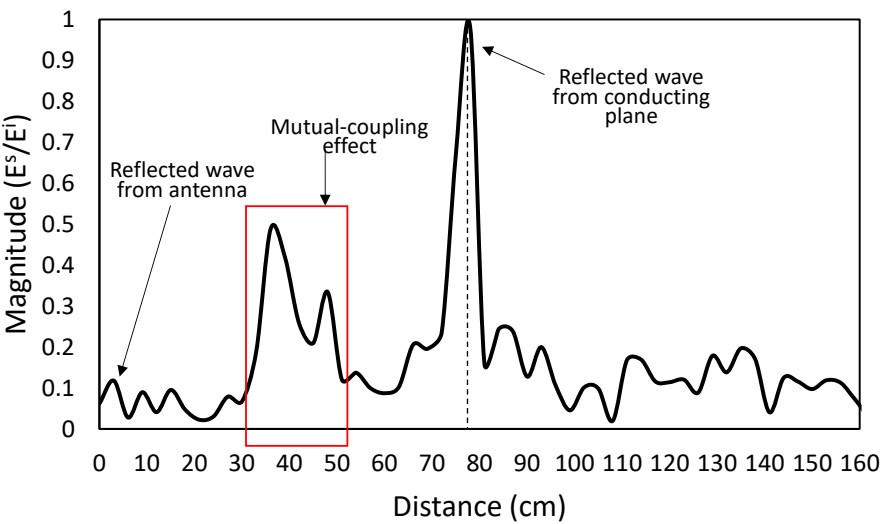

**Figure 8.** Time-domain response of the scattered wave from the conducting plane.

Figure 8 depicts the magnitude of the scattered wave to the receiving antenna in time-domain where the position of the peak represents the distance from the phase center of scattered wave to the receiving port of the network analyzer. For instance, the first peak that appeared at 2.5 cm represented the reflected wave from the receiving antenna. It was also the distance from phase center of the antenna to the receiving port of the network analyzer. The highest peak that took place at 78 cm represented the time-domain response of the backscattered wave from the conducting plane. The 78 cm distance could be considered as the summation of distance between transmitting port and receiving port of the network analyzer to the conducting plane—$2R = 60$ cm. The distance of 18 cm was for the wave travelling through the amplifier. Some peaks, that appeared in the distance of 30–50 cm, were the response of mutual coupling between the antennas. Although its magnitude was half of the scattered wave from the conducting plane, the magnitude was significantly greater than the other backscattered wave components from the LCS.

The next two subsections show the time-domain response of the scattered wave from the LCS 3 and LCS 4. The distance between the ports of the network analyzer and the surface of the LCS was 30 cm, $R = 30$ cm. Therefore, the response of the reflected wave $(F_1)$ from the surface of the LCS appeared at 78 cm as it was the same position of the response of the reflected wave from the conducting plane. The responses of other backscattered wave components were beyond 78 cm. Therefore, a gating of time-domain response was selected to display the measurement results without the effect of mutual coupling.

### 3.4. LCS 3 $(r_1 = 3\ cm, r_2 = 2.5\ cm, \varepsilon_{r_0} = 1, \varepsilon_{r_1} = 3.9 + 0.12j, \varepsilon_{r_2} = 7.52 + 3.93j)$

The core of LCS 3 was constructed from a 5 cm-diameter hollow plastic ball which was filled with syrup. It was covered with a 0.5 cm-thick plasticine. The dielectric properties of the plasticine and the syrup were measured at 10 GHz by the FieldFox network analyzer and an 85070E Agilent Dielectric Probe Kit [31]. The corresponding dielectric properties were respectively $\varepsilon_{r_1} = 3.9 + 0.12j$ and $\varepsilon_{r_2} = 7.52 + 3.93j$. The time-domain response (solid line) of the backscattered wave from LCS 3 is shown in Figure 9. The magnitude and phase of the backscattered wave (dot) from LCS 3 was calculated by the principle presented in the previous section and plotted on the same graph.

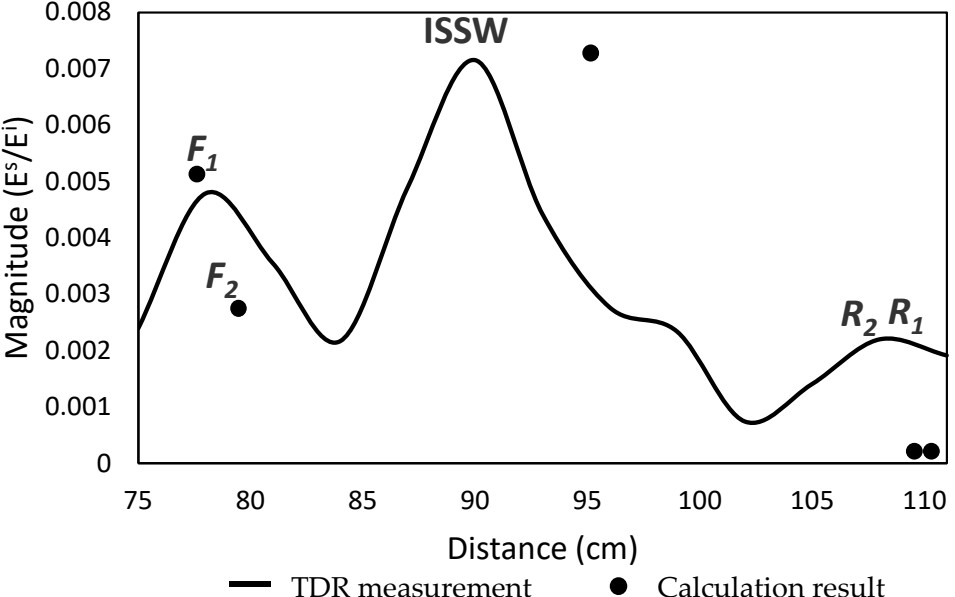

**Figure 9.** Time-domain response of the backscattered wave from LCS 3.

Figure 9 shows two peaks at 78 cm and 90 cm corresponding to the time-domain response of the $F_1$ and ISSW waves. Similarly, the $R_1$ and $R_2$ appear around 110 cm. They agreed well with the time delay (presented in distance) among these calculated components. The discrepancy in distance was from the velocity calculated in free space while the actual wave propagated in the dielectric materials. The time-domain response of the backscattered wave $(F_2)$ disappeared in Figure 9 because the distance between $F_2$ compared to $F_1$ was 1.9 cm while the resolution of the measurement system was 3.75 cm. The agreement between the measurement result and calculation result indicated three main contributors $(F_1, F_2,$ and ISSW) of the backscattered wave from LCS3. The magnitude of the calculated $F_1$ component was 3.4% higher than the measurement result. In addition, the difference of distance of ISSW component between calculation and measurement results was 5.1%. This depicted that the measurement supported the wave component calculations.

### 3.5. LCS 4 $(r_1 = 4\ cm, r_2 = 3\ cm, \varepsilon_{r_0} = 1, \varepsilon_{r_1} = 7.52 + 3.93j, \varepsilon_{r_2} = 21.4 + 19.2j)$

The LCS 4 was constructed from two hollow plastic balls that were fixed with the same origin. The 6 cm-diameter core was filled with the liquid made of syrup and sterile

water while the 1 cm-thick shield of LCS 4 was filled with syrup. The dielectric properties of the core and the shield were determined at 10 GHz with a dielectric probe and were $\varepsilon_{r_1} = 3.9 + 0.12j$, $\varepsilon_{r_2} = 7.52 + 3.93j$, respectively. A similar measurement procedure for LCS 3 was repeated for LCS 4 and the backscattered wave components from LCS 4 is presented in Figure 10. In Figure 10, three peaks obviously appeared at 78 cm, 90 cm, and 100 cm corresponding to the time-domain response of the $F_1$, $F_2$, and ISSW components. The highest peak belonged to the $F_1$ component instead of the ISSW as in the case of LCS 3 due to the higher lossy materials of both layers (shield and core) of LCS 4. The position of the $F_1$ component matched well with the calculation results whereas that for ISSW had a slight difference due to the limited resolution of the measurement system. With the thicker shield and higher dielectric properties of the shield of LCS 4 compared to the one of LCS 3, from the calculation, the wave of $F_2$ was 5.7 cm behind the wave of $F_1$ and the time-domain response of $F_2$ was present. The magnitude difference of $F_1$ component was 6.3% whereas the distance difference for ISSW was 6.1%. The agreement between the measurement and calculation results indicated two main contributors ($F_1$ and ISSW) of the backscattered wave from LCS 4. It should be pointed out that the measurement results in Figures 9 and 10 verify the wave components calculated by the proposed models.

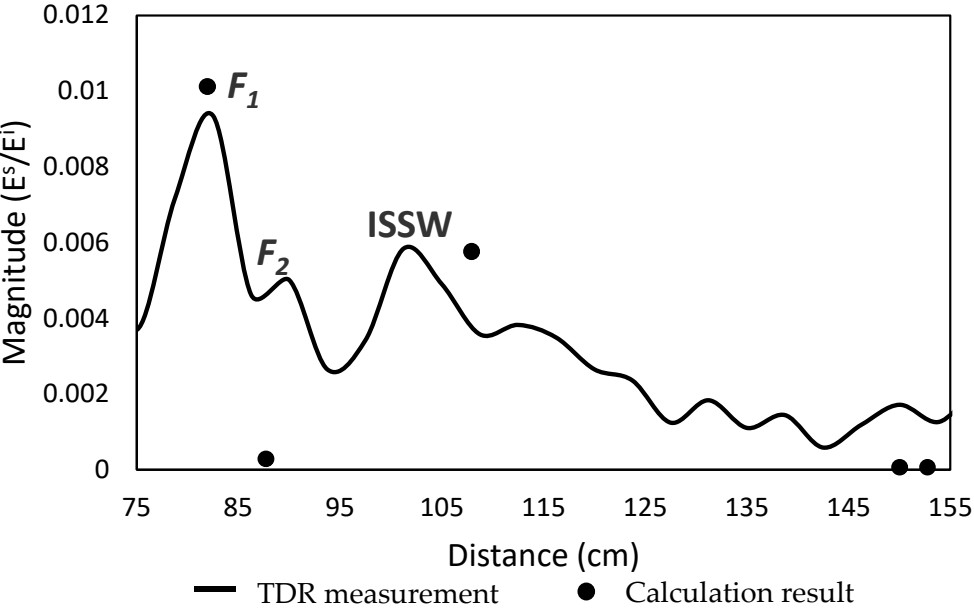

**Figure 10.** Time-domain response of the backscattered wave from LCS 4.

## 4. Application in Fruit Characterization

To illustrate the usefulness of the proposed models, this section shows an application of the proposed models in fruit characterization. Since mangosteen is widely cultivated due to its unique sweet–sour taste and has an economic impact, this section demonstrates the dielectric properties determination of mangosteen. Generally, the normal flesh of mangosteen has a good taste, and it can be recognized by the white color of flesh as shown in Figure 11a. However, according to an excessive amount of water during mangosteen's development, its flesh becomes translucent as shown in Figure 11b. The translucent flesh is the main contributor to internal defects and is hard to detect non-destructively. Some research works related to non-destructive mangosteen grading have been published in [32–36] but they did not determine dielectric properties from measured scattered wave which could be suitable for practical fruit classification. To demonstrate determination of dielectric properties of mangosteen, 15 mangosteens, of export grade, were collected. Although mangosteen fruits have a spherical shape, the diameters in different positions are different. The dimensions of the mangosteen fruits were measured as seen in Figure 12a–c and the diameters were denoted as shown in Figure 12a as $D_1$, $D_2$, and $D_3$. They were

related to radius of inner and outer spheres in the calculation models. The respective values of $D_1$, $D_2$, and $D_3$ were $61.1 \pm 4.8$ mm, $59.2 \pm 4.9$ mm, and $52.7 \pm 4.8$ mm. The thickness of peel of mangosteen fruits were measured as seen in Figure 12b,c. The values of $T_1$, $T_2$, $T_3$ were $8.8 \pm 2$ mm, $6.8 \pm 2$ mm, and $10.9 \pm 1$ mm. The thickness corresponds to the difference of spherical radii in the models in Section 2. The diameter and the thickness were averaged for the parameters in calculation. The measurement setup was same as the one in the previous section, shown in Figure 7. The dielectric spheres in Figure 7c,d (plasticine and syrup-filled spherical plastic ball) were replaced by mangosteen fruits. In addition, instead of wideband measurement, the narrowband frequency of 10 GHz was fixed.

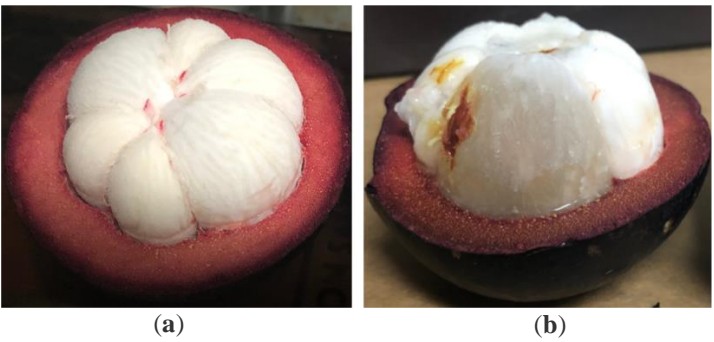

(a)　　　　　　　　　　(b)

**Figure 11.** Mangosteen flesh; (**a**) Normal flesh; (**b**) Translucent flesh.

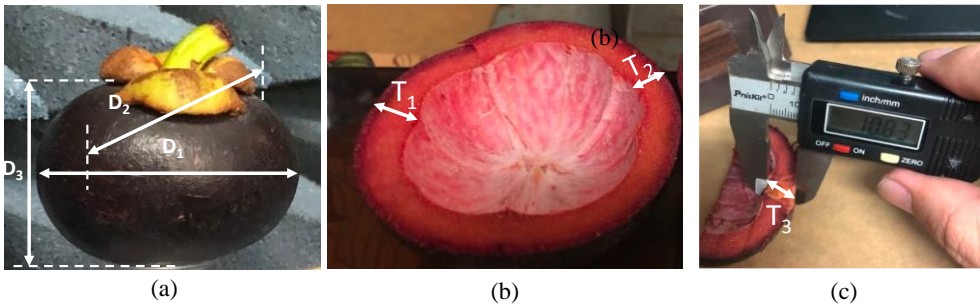

(a)　　　　　　　　　　(b)　　　　　　　　　　(c)

**Figure 12.** Physical dimensions of mangosteen fruit; (**a**) Outer diameter measurement; (**b**) Thickness of peel around the fruit; (**c**) Thickness of peel at the bottom side of the fruit.

The dielectric properties of mangosteen fruits were measured with the dielectric probe. It was found that the translucent flesh had $\varepsilon'_r$ and $\varepsilon''_r$ of $46.9 \pm 1$ and $28.1 \pm 0.2$, respectively. The corresponding values for normal flesh were $41.1 \pm 4$ and $25.6 \pm 2.5$, respectively. For peel, the values of $\varepsilon'_r$ and $\varepsilon''_r$ were quite constant at 6.4 and 2.1.

The dielectric constant, $\varepsilon'_r$, and loss factor, $\varepsilon''_r$, of the translucent flesh were higher than the one of the normal flesh. The large variation of the dielectric constant, $\Delta\varepsilon'_r = 5.8$ and the loss factor, $\Delta\varepsilon''_r = 2.5$ were observed. Therefore, the dielectric properties of flesh of mangosteen could be used as a non-destructive indicator to detect the internal defect of mangosteen with microwave. The attenuation constant and phase constant of wave propagating inside flesh were from 30.26 to 30.52 dB/cm and 12.80 to 13.54 degree/cm whereas those for peel were 7.45 dB/cm and 5.37 degree/cm. With a high loss factor in peel and a very high loss factor in flesh, mangosteen was a high lossy medium. Hence, Glory wave and internal surface scattering wave were negligibly small.

The uniform plane wave illuminated at the bottom of the mangosteen fruit. From the physical dimensions and the dielectric properties of mangosteen, the backscattered wave from the mangosteen model consisted of the first front axial return wave ($F_1$), the second front axial return wave ($F_2$), the first rear axial return wave ($R_1$), the second rear axial return wave ($R_2$), and the internal surface scattering wave ($I_2$). As the fruit was a lossy dielectric, hence the wave components ($R_1$, $R_2$, and $I_2$) that propagated through flesh were attenuated and the magnitude of such waves were neglected. Therefore, the total backscattered wave

consisted of $F_1$ and $F_2$. The diameter of mangosteen, in this experiment, was measured manually by using a ruler. Since dielectric properties of peel were quite constant, it was assumed to be a known parameter. The field of $F_1$ was calculated by (4) while the field of $F_2$ was the complex subtraction the field of $F_1$ from the total backscattered wave. With the measured magnitude and phase of $F_2$, the reflection coefficient between peel and flesh $\Gamma_{12} = |\Gamma_{12}|e^{j\varphi_2}$ was calculated by (5). Here we utilized the dielectric properties of peel $\varepsilon_{r1} = 6.4 + 2.1j$ and thickness of peel of 11 mm from the averaged value of the measurement results. The dielectric constant and conductivity of flesh could be found by equating the real part and imaginary part of the relationship between intrinsic impedance and reflection coefficient of wave reflected from flesh, respectively. Hence, the dielectric constant and conductivity of flesh could be found from (21) and (22), respectively.

$$\varepsilon'_{r2} = \frac{|\Gamma_{12}|^4 - 2|\Gamma_{12}|^2\left(1 + 2\sin^2\varphi_2\right) + 1}{\left(|\Gamma_{12}|^2 + 2|\Gamma_{12}|\cos\varphi_2 + 1\right)^2}\varepsilon'_{r1} - \frac{4|\Gamma_{12}|\sin\varphi_2\left(1 - |\Gamma_{12}|^2\right)}{\left(|\Gamma_{12}|^2 + 2|\Gamma_{12}|\cos\varphi_2 + 1\right)^2}\sigma_1\frac{\eta_0^2}{\omega\mu_0} \quad (21)$$

$$\sigma_2 = \frac{|\Gamma_{12}|^4 - 2|\Gamma_{12}|^2\left(1 + 2\sin^2\varphi_2\right) + 1}{\left(|\Gamma_{12}|^2 + 2|\Gamma_{12}|\cos\varphi_2 + 1\right)^2}\sigma_1 + \frac{4|\Gamma_{12}|\sin\varphi_2\left(1 - |\Gamma_{12}|^2\right)}{\left(|\Gamma_{12}|^2 + 2|\Gamma_{12}|\cos\varphi_2 + 1\right)^2}\varepsilon_{r1}\frac{\omega\mu}{\eta_0^2} \quad (22)$$

The results obtained from a dielectric probe and a network analyzer measurement, after backscattered wave measurement, were used as predicted values. The measured results from the scattered wave measurement along with the calculation with the proposed models were shown as the measured values. They were plotted in Figure 13 where Figure 13a shows the results for dielectric constant and Figure 13b is for conductivity.

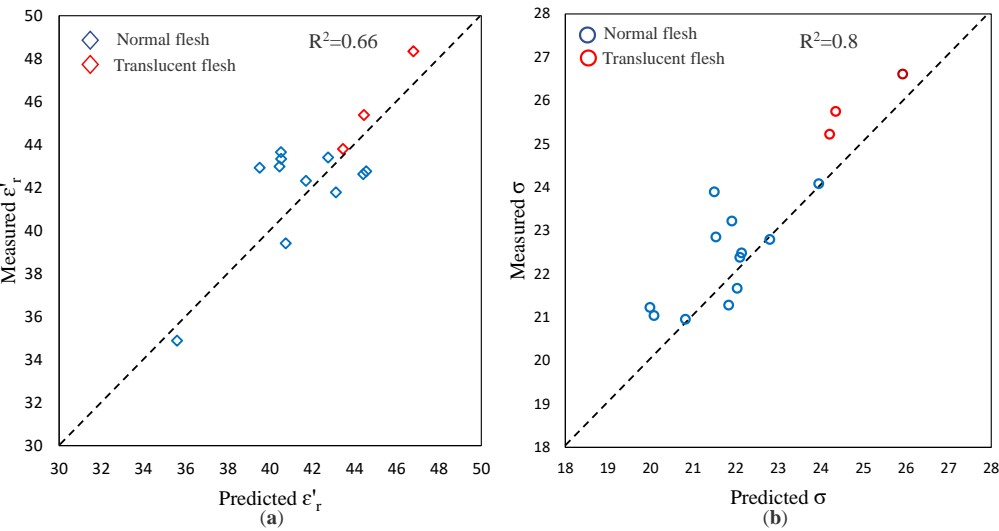

**Figure 13.** Comparison of the dielectric properties of flesh determined from the backscattered wave measurement and those from a dielectric probe: (**a**) dielectric constant; (**b**) conductivity.

The number of samples was fifteen and the normal samples were shown in "blue" while the translucent samples were in "red". The $R^2$ for dielectric constant and conductivity were 0.66 and 0.8, respectively. This exhibited the good agreement between the predicted and measured results by the scattered wave measurement. The variation could be attributed from the variation of size and thickness of peel from the set averaged values. With the appropriate threshold for the dielectric constant, the high accuracy could be achieved. In this demonstration, for the threshold of dielectric constant of 44.5, a grading accuracy of 93.33% could be achieved. It should be noted that the separation between the conductivity of translucent flesh and normal flesh is more obvious than that for dielectric constant.

Hence, the accuracy is much increased. With the two indicators from the dielectric constant and conductivity, the grading accuracy can be further improved.

The comparison of the performance of mangosteen grading techniques are depicted in Table 1. The work in [32] used a probe to sense moisture content of mangosteen flesh which is higher than that of normal one. With the suitable threshold of magnitude of reflected wave, the accuracy of 79% was achieved. The limitation of this technique is that the probe must be contact with the fruit. Hence, it is not suitable for grading large number of fruits in a continuous process.

**Table 1.** Comparison of performance of mangosteen fruit grading techniques.

| References | Technique | Fruit | Accuracy (%) |
|---|---|---|---|
| [32] | Microwave probe | Mangosteen | 79.0 |
| [33] | Physical and chemical | Mangosteen | 78.9 |
| [34] | Color | Mangosteen | 67.4 |
| [35] | Strain gage | Mangosteen | 78.57 |
| [36] | Vis/NIR | Mangosteen | 92.92 |
| This work | Scattered wave measurement | Mangosteen | 93.33 |

The work in [33] shows that physical and chemical parameters of mangosteen samples were determined as a ratio of maximum diameter to minimum diameter. Discrimination analyses were performed on the parameters to evaluate the accuracy of translucency classification. The overall accuracy of classification was achieved using all parameters presenting 78.9%. The work in [34] presented a method for predicting damage based on color of the stem of the mangosteen. The accuracy of predicting internal defects from the color variation of two spots on the surface of the same fruit. The percentage of accurate prediction was 67.4%. The work in [35] proposed the variation frequency based on strain gage sensor to predict an internal translucent and yellow gummy latex in mangosteen fruits. The measurements were performed by vibrating the frequency of 25, 30, 35, and 40 Hz. The evaluation of feature extraction based on time and frequency domain provided accuracy of 78.57%. This technique needs measurement for many days and some parameters such as hardening pericarp, fruit size, and skin color must be rejected before evaluation. The work in [36] shows the possibility to develop a non-destructive technique using Vis/NIR reflectance spectroscopy for measuring internal quality intact mangosteen fruit. Good classification could be achieved with an accuracy of 92.92% at the expense of whole region of wavelengths. The appropriate wavelength must be found to obtain a cost-effective sensor. Among various techniques, the Vis/NIR possessed the highest accuracy, but the sensor was expensive. The customized sensor, where only the narrow band was suitable for this application, can be attractive. The results presented in this paper exhibited a good candidate for mangosteen classification since a cost-effective narrow band microwave reflectometer can be realized. The accuracy based on 15 samples could be accomplished with the accuracy of 93.33%. The experiment with a large size of samples will be performed in future work.

## 5. Conclusions

The model for determining various components of backscattered wave from a lossy concentric dielectric sphere enables one to understand the insight behavior of many realistic objects possessing lossy concentric dielectric sphere structure. According to a single bounce assumption in each component, the calculation results were accurate for lossy dielectric. The calculation accuracy was validated by comparing the results from the proposed model with the one calculated using commercial software and experimental results agree well for the case of lossy material. With the detail of the backscattered wave components, this model could be applied for the determination of dielectric properties of both layers of the material. The future work will be an application of this model in fruit classification sensor.

**Author Contributions:** Conceptualization, H.N.D., C.P. and M.K.; methodology, H.N.D., C.P. and M.K.; software, H.N.D.; validation, H.N.D. and M.K.; formal analysis, H.N.D.; investigation, H.N.D.; resources, M.K.; data curation, H.N.D.; writing—original draft preparation, H.N.D.; writing—review and editing, C.P. and M.K.; visualization, H.N.D. and M.K.; supervision, M.K.; project administration, H.N.D.; funding acquisition, M.K. All authors have read and agreed to the published version of the manuscript.

**Funding:** This research was funded by King Mongkut's Institute of Technology Ladkrabang under grant no. KDS2019/001.

**Conflicts of Interest:** The authors declare no conflict of interest.

**Appendix A**

Transmission and reflection coefficients for the uniform plane wave propagates from the $m^{th}$ medium to the $l^{th}$ medium $[(m, l) = (1, 0), (0, 1), (1, 2), (2, 1)]$ with the oblique incidence wave ($\theta_i$, $\theta_t$ are the incident angle and refraction angle, respectively) [30]. Note that the normal incident wave, the incident, and refraction angles are equal as $\theta_i = \theta_t = 0°$.

Perpendicular polarization

$$|T_{\perp ml}| = \frac{2|\eta_l| \cos \theta_i}{\sqrt{\left(|\eta_l| \cos \theta_i \cos \varphi_{\eta_l} + |\eta_m| \cos \theta_t \cos \varphi_{\eta_m}\right)^2 + \left(|\eta_l| \cos \theta_i \sin \varphi_{\eta_l} + |\eta_m| \cos \theta_t \sin \varphi_{\eta_m}\right)^2}} \tag{A1}$$

$$\varphi_{T_{\perp ml}} = \varphi_{\eta_l} - \tan^{-1}\left(\frac{|\eta_l| \cos \theta_i \sin \varphi_{\eta_l} + |\eta_m| \cos \theta_t \sin \varphi_{\eta_m}}{|\eta_l| \cos \theta_i \cos \varphi_{\eta_l} + |\eta_m| \cos \theta_t \cos \varphi_{\eta m}}\right) \tag{A2}$$

$$|\Gamma_{\perp ml}| = \sqrt{\frac{\left(|\eta_l| \cos \theta_i \cos \varphi_{\eta_l} - |\eta_m| \cos \theta_t \cos \varphi_{\eta_m}\right)^2 + \left(|\eta_l| \cos \theta_i \sin \varphi_{\eta_l} - |\eta_m| \cos \theta_t \sin \varphi_{\eta_m}\right)^2}{\left(|\eta_l| \cos \theta_i \cos \varphi_{\eta_l} + |\eta_m| \cos \theta_t \cos \varphi_{\eta_m}\right)^2 + \left(|\eta_l| \cos \theta_i \sin \varphi_{\eta_l} + |\eta_m| \cos \theta_t \sin \varphi_{\eta_m}\right)^2}} \tag{A3}$$

$$\varphi_{\Gamma_{\perp ml}} = \tan^{-1}\left(\frac{|\eta_l| \cos \theta_i \sin \varphi_{\eta_l} - |\eta_m| \cos \theta_t \sin \varphi_{\eta_m}}{|\eta_l| \cos \theta_i \cos \varphi_{\eta_l} - |\eta_m| \cos \theta_t \cos \varphi_{\eta_m}}\right) - \tan^{-1}\left(\frac{|\eta_l| \cos \theta_i \sin \varphi_{\eta_l} + |\eta_m| \cos \theta_t \sin \varphi_{\eta_m}}{|\eta_l| \cos \theta_i \cos \varphi_{\eta_l} + |\eta_m| \cos \theta_t \cos \varphi_{\eta_m}}\right) \tag{A4}$$

Parallel polarization

$$\left|T_{\| ml}\right| = \frac{2|\eta_l| \cos \theta_i}{\sqrt{\left(|\eta_m| \cos \theta_i \cos \varphi_{\eta_m} + |\eta_l| \cos \theta_t \cos \varphi_{\eta_l}\right)^2 + \left(|\eta_m| \cos \theta_i \sin \varphi_{\eta_m} + |\eta_l| \cos \theta_t \sin \varphi_{\eta_l}\right)^2}} \tag{A5}$$

$$\varphi_{\Gamma_{\| ml}} = \varphi_{\eta_l} - \tan^{-1}\left(\frac{|\eta_m| \cos \theta_i \sin \varphi_{\eta_m} + |\eta_l| \cos \theta_t \sin \varphi_{\eta_l}}{|\eta_m| \cos \theta_i \cos \varphi_{\eta_m} + |\eta_l| \cos \theta_t \cos \varphi_{\eta_l}}\right) \tag{A6}$$

$$\left|\Gamma_{\| ml}\right| = \sqrt{\frac{\left(-|\eta_m| \cos \theta_i \cos \varphi_{\eta_m} + |\eta_l| \cos \theta_t \cos \varphi_{\eta_l}\right)^2 + \left(-|\eta_m| \cos \theta_i \sin \varphi_{\eta_m} + |\eta_l| \cos \theta_t \sin \varphi_{\eta_l}\right)^2}{\left(|\eta_m| \cos \theta_i \cos \varphi_{\eta_m} + |\eta_l| \cos \theta_t \cos \varphi_{\eta_l}\right)^2 + \left(|\eta_m| \cos \theta_i \sin \varphi_{\eta_m} + |\eta_l| \cos \theta_t \sin \varphi_{\eta_l}\right)^2}} \tag{A7}$$

$$\varphi_{\Gamma_{\| ml}} = \tan^{-1}\left(\frac{-|\eta_m| \cos \theta_i \sin \varphi_{\eta_m} + |\eta_l| \cos \theta_t \sin \varphi_{\eta_l}}{-|\eta_m| \cos \theta_i \cos \varphi_{\eta_m} + |\eta_l| \cos \theta_t \cos \varphi_{\eta_l}}\right) - \tan^{-1}\left(\frac{|\eta_m| \cos \theta_i \sin \varphi_{\eta_m} + |\eta_l| \cos \theta_t \sin \varphi_{\eta_l}}{|\eta_m| \cos \theta_i \cos \varphi_{\eta_m} + |\eta_l| \cos \theta_t \cos \varphi_{\eta_l}}\right) \tag{A8}$$

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
