# Peer review of "Approximated Backscattered Wave Models of a Lossy Concentric Dielectric Sphere for Fruit Characterization"

_electronics, doi:10.3390/electronics11101521_

Round 1
Reviewer 1 Report
It took me long to review this manuscript because many equations and derivation are involved. I am not 100% sure about every calculation and I would suggest the authors proofread these equations to avoid any error. I have few questions as follows.
1. Can you please provide comparative results from the calculation and measurement? Otherwise so many equations and calculation would make no sense.
2. As put in the title, the models were built for fruit characterization. I would like to see experimental result using real fruit instead of any substitute. Is it fine to supplement such experiments?
3. Please compare the experimental result of this work with the state of the art and present the technical advances clearly (e.g., in a table).
Author Response
Dear Reviewer.
We faithfully appreciate your comments and questions which improves our manuscript significantly. We did our best to respond your comments and hope it will be satisfactory. The responses to your comments are listed as follows.
- Can you please provide comparative results from the calculation and measurement? Otherwise so many equations and calculation would make no sense.
Response: Comparative results from the calculation and measurement are divided into two parts.
- Section 3.4 (Yellow highlight in lines 410-413 of page 14) and in section 3.5 (Yellow highlight in lines 431 to 433 of page 15) compared the results from Time domain measurement with the calculation results.
- Section 4 is added. We compared the dielectric properties determined from scattered wave measurement and calculated by our proposed models with the measurement results using a dielectric probe and a vector network analyzer. The results are displayed in Fig.13 of page 17. The discussion on this result is below Fig.13.
As put in the title, the models were built for fruit characterization. I would like to see experimental result using real fruit instead of any substitute. Is it fine to supplement such experiments?
Response: Section 4 is added. It shows the detail how we setup experiments with mangosteen fruits. The results are displayed in this section.
Please compare the experimental result of this work with the state of the art and present the technical advances clearly (e.g., in a table).
Response: Comparison the experimental result of this work with the state of the art is listed in Table 1 of section 4. The table is in page 18 and the discussion is depicted below the table in page 18.

Reviewer 2 Report
The paper proposes that the dielectric properties of a concentric dielectric sphere object (e.g. fruit) could be estimated from some wave components of the backscattered wave. In this sense, the total scattered field results and the reported values agreed with the values calculated by the wave functions presented. In general the paper has potential quality and interest and in that sense, adding a research question is recommended.
Author Response
Dear Reviewer.
We faithfully appreciate your comments and questions which improves our manuscript significantly. We did our best to respond your comments and hope it will be satisfactory. The responses to your comments are listed as follows.
The paper proposes that the dielectric properties of a concentric dielectric sphere object (e.g. fruit) could be estimated from some wave components of the backscattered wave. In this sense, the total scattered field results and the reported values agreed with the values calculated by the wave functions presented. In general the paper has potential quality and interest and in that sense, adding a research question is recommended.
Response: Research question is added in section 1, page 1. It is Yellow highlighted in lines 28-37.

Reviewer 3 Report
Without questioning the relevance of the study, it is necessary to note a number of drawbacks:
1. It is not clear from section 2 what was proposed and what consists the novelty of the research. It is necessary to clearly separate the proposed methods from the existing ones.
2. The authors state in the title of the manuscript a method for the characteristics of the fruits. However, the paper proposes only a method for solving the direct model problem of computing the reflected wave from a sphere with known parameters. Undoubtedly, the considered model can be used to characterize fruits, but in order to achieve it one also need to solve the inverse problem. Namely, the determination of the parameters of the sphere by the reflected wave. Experiments were also carried out only on a kind of ideal spheres, and not on fruits. Thus, the problem stated in the title of the manuscript has not been solved.
3. I am not sure if the word "model" is correctly used in section 3 to describe specific parameters of spheres. Confusion with the models from section 2 is present.
4. A comparison with accurate analytical solutions, for example with Mie solution, is required.
5. For greater openness and reproducibility of the results obtained, I strictly recommend to publish the source code of the proposed method in an open repository.
Author Response
Dear Reviewer.
We faithfully appreciate your comments and questions which improves our manuscript significantly. We did our best to respond your comments and hope it will be satisfactory. The responses to your comments are listed as follows.
- It is not clear from section 2 what was proposed and what consists the novelty of the research. It is necessary to clearly separate the proposed methods from the existing ones.
Response: Thank you for your valuable comment. The novelty is claimed in section 1.
It is highlighted in Yellow in Lines 76-84 of page 2.
- The authors state in the title of the manuscript a method for the characteristics of the fruits. However, the paper proposes only a method for solving the direct model problem of computing the reflected wave from a sphere with known parameters. Undoubtedly, the considered model can be used to characterize fruits, but in order to achieve it one also need to solve the inverse problem. Namely, the determination of the parameters of the sphere by the reflected wave. Experiments were also carried out only on a kind of ideal spheres, and not on fruits. Thus, the problem stated in the title of the manuscript has not been solved.
Response: Thank you very much for this comment. We agree that it is important to show the inverse problem with real fruit in this paper. So section 4 is added in pages 15-18. This section shows the inverse problem for mangosteen fruits characterization. The principle was explained and followed by experiment setup and results.
- I am not sure if the word "model" is correctly used in section 3 to describe specific parameters of spheres. Confusion with the models from section 2 is present.
Response: Thank you for your observation. The word “model” in section 3 was replaced by “LCS”.
- A comparison with accurate analytical solutions, for example with Mie solution, is required.
Response: We tried to compare our results with analytic solutions such as Mie solution
but could not carry out in this time. Therefore, comparison was done with a commercial
software CST Studio which, from our experience, is acceptable accurate.
The results slightly change from calculation with MATLAB in the previous submission.
- For greater openness and reproducibility of the results obtained, I strictly recommend to publish the source code of the proposed method in an open repository.
Response: We attached the excel program, we developed for calculation, with this response. There are six files to be used together.
Please see the guideline to operate it.
Round 2
Reviewer 3 Report
The manuscript has become much soundness after revision and can be published in Electronics.